# Phase Transformations of 5Cr-0.5Mo-0.1C Steel after Heat Treatment and Isothermal Exposure

Maribel L. Saucedo-Muñoz [1,*], Victor M. Lopez-Hirata [1], Hector J. Dorantes-Rosales [1], Jose D. Villegas-Cardenas [2], Diego I. Rivas-Lopez [1], Manuel Beltran-Zuñiga [1], Carlos Ferreira-Palma [3] and Joel Moreno-Palmerin [4]

1   Instituto Politecnico Nacional (ESIQIE), Ciudad de Mexico 07300, Mexico
2   Universidad Politecnica del Valle de Mexico, Tultitlán 54900, Mexico
3   Universidad Veracruzana (FCQ), Boca del Río 94294, Mexico
4   Departamento de Ingeniería en Minas, Metalurgia y Geología, Universidad de Guanajuato, Guanajuato 36000, Mexico
*   Correspondence: msaucedom@ipn.mx; Tel.: +525-557-296-000 (ext. 54206)

**Abstract:** This study consists of the experimental and numerical analysis of the phase transformations of 5Cr-0.5Mo.0.1C steel after heat treatment. The microstructure of the as-received steel comprised ferrite and bainite, which is in agreement with the microconstituents predicted by the Calphad-calculated TTT diagram. Calphad-based precipitation calculations show that the cooling stage of normalizing treatment did not cause carbide formation. In contrast, tempering at 700 °C for 15 min promotes the intergranular precipitation of $Fe_3C$, $M_7C_3$ and $M_{23}C_6$ carbides, which is consistent with experimental results. Aging at 600 °C for short periods caused the precipitation of both $M_7C_3$ and $M_{23}C_6$ carbides; however, $M_{23}C_6$ is the dominant phase after prolonged aging. This is in agreement with experimental results. A rapid decrease in the steel hardness was observed after short aging, which is attributable to bainite transformation. Further reduction in hardness is associated with the diffusion-controlled coarsening of $M_{23}C_6$ carbide.

**Keywords:** C-Cr-Mo steels; phase transformations; precipitation; heat treatment; mechanical properties



## 1. Introduction

Low-alloyed ferritic steels are widely used for the fabrication of equipment for petroleum refineries and carbon power plants [1,2]. This type of steel exhibits good mechanical properties, including good tensile strength and toughness at room temperature and creep resistance at temperatures up to 600 °C. This type of steel is designated as heat-resistant steel [3]. The low-C Cr-Mo ferritic steels are an example of heat-resistant steels. Higher Cr and Mo contents increase the mechanical strength at high temperatures [4]. Besides solid solution strengthening, these steels also present carbide precipitation hardening. The precipitation hardening treatment of low-carbon 1Cr-0.5 Mo steel relies on the formation of $M_3C$, $M_7C_3$ and $M_{23}C_6$ carbides [5]. In contrast, the precipitation hardening of low-carbon 2.25Cr-1Mo steel originates from the presence of $M_{23}C_6$ and $Mo_2C$ [6]. The precipitation of carbides is relevant for improving the creep strength, since the alloyed carbides constitute barriers that impede dislocation movement and grain boundary sliding during creep at high temperatures [7,8]. The ferritic low-carbon 5Cr-0.5 Mo also presents good creep strength at temperatures up to 600 °C [8]. These characteristics are achieved by heat treatments and precipitation hardening, which contribute to maintaining their properties at high temperatures. There are few studies on phase transformations during heat treatments and heating at operating temperatures. Therefore, it is relevant to carry out this type of study to better understand the relationship between the microstructure and properties of this steel.

The heat treatment of these steels consists of normalizing and tempering, which allows achieving the required tensile strength and promotes the formation of precipitates to present good creep strength during service operation [1].

Calphad-based software [8–13] is beneficial for understanding the phase stability and the precipitation of carbides during the isothermal exposure of low-alloyed steels. This software allows us to estimate the growth kinetics of precipitation and coarsening of precipitates, which are important in the deterioration of creep properties during its service operation at high temperatures in industrial components.

We performed numerical and experimental analyses of the phase transformations after heat treating and the precipitation process after isothermal aging at 600 °C for a low carbon 5Cr-0.5 Mo-0.1C steel to understand their effect on mechanical properties.

## 2. Materials and Methods

### 2.1. Numerical Method

With the Calphad-based tools, Thermo-Calc (TC) and TC-Prisma [11–16], we established the phase stability using the steel composition of Table 1. The time–temperature–transformation (TTT) diagram and precipitation growth kinetics during the heat-treatment process and isothermal aging at 600 °C were also calculated. The TCFe11 and MobFe6 databases (Thermo-Calc, Stockholm, Sweden) were employed [17], assuming an austenite grain size of 30 μm and grain boundary precipitation. This software was also used to determine the interfacial free energy of the precipitate/matrix.

**Table 1.** Chemical composition (wt %) of the present work's steel.

| C | Cr | Mo | Mn | Si | Cu | Ni | Ti |
|---|---|---|---|---|---|---|---|
| 0.10 | 4.50 | 0.46 | 0.36 | 0.35 | 0.05 | 0.12 | 0.002 |

### 2.2. Experimental Method

Table 1 presents the chemical composition of the steel. Specimens were cut to 10 mm × 10 mm × 10 mm dimensions. The initial condition of the steel, referred to as "*as-received*", was obtained after normalizing at 950 °C for 30 min and tempering at 715 °C for 15 min. Additionally, samples were aged at 600 °C for periods between 0 and 7000 h. We used a scanning electron microscope (SEM, JEOL Ltd., Tokyo, Japan) equipped with EDX for the microstructural characterization of as-received and aged samples. Electrolytic dissolution of the ferrite phase with 2 vol % $HNO_3$ in $CH_3$-OH at 4 V (dc) enabled the extraction of the precipitates from the ferrite matrix. The residues were collected, rinsed with water and subsequently dried. The X-ray diffraction (XRD, JEOL LTD., Tokyo, Japan) analysis of these residues with a monochromated Cu K$\alpha$ radiation allowed us to determine the carbide type. These residues were mounted on carbon-coated 300 mesh copper grits, and subsequently analyzed by TEM equipped with EDX at 200 kV. Rockwell "B" hardness of aged specimens was measured according to the ASTM E-18 standard [18]. Grain size was determined using the intercept procedure according to the ASTM E-112 standard [19]. The volume fraction of precipitates was quantified using SEM micrographs at 30,000×, based on a point-counting method consistent with ASTM E-562 standard [20].

## 3. Results

### 3.1. Phase Stability and TTT Diagram

Figure 1 shows the variation in the volume fraction of all equilibrium phases against temperature, calculated by Calphad-based software [17]. This figure indicates the formation of austenite from the liquid during cooling. TiC precipitates from the austenite matrix at temperatures below 915 °C. This carbide also contains a small amount of Cr and Mo. These precipitates contribute to control grain size, as presented in subsequent sections. As temperature decreases, austenite transforms into ferrite around 827 °C, the $A_3$ intercritical

temperature. These calculations also suggest that the precipitation of Cr-rich $M_{23}C_6$ is possible by aging at temperatures below approximately 810 °C.

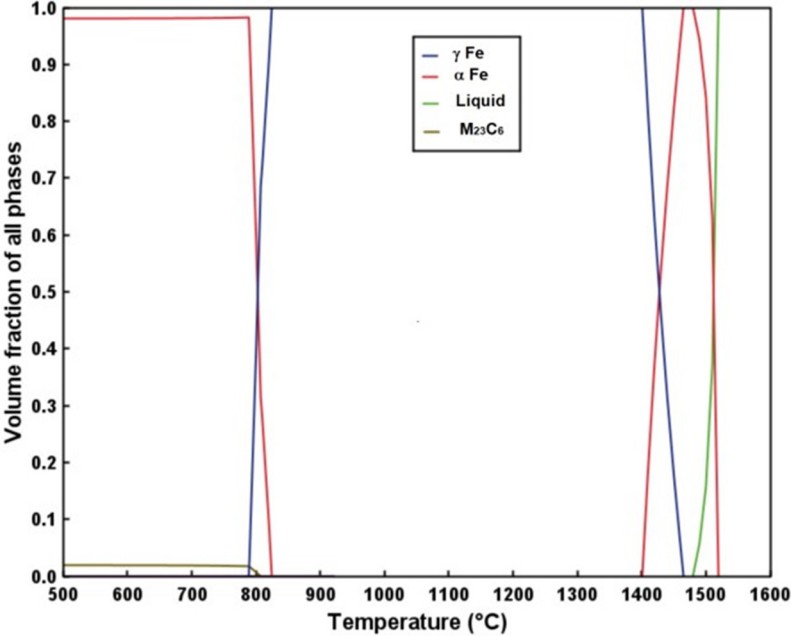

**Figure 1.** Plot of volume fraction of equilibrium phases against temperature.

The time–temperature–transformation diagram, TTT, calculated by a Calphad-based steel transformation software [17], is shown in Figure 2. The precipitation of ferrite from austenite below the $A_3$ temperature was determined using Calphad-based precipitation software [17]. This diagram suggests that the microstructure in the normalized condition, at a cooling rate close to 1 °C/s, consists of ferrite and bainite. This means that the first phase transformation expected to occur during the cooling is the ferrite precipitation in the austenite matrix at temperatures lower than 820 °C (Figure 1). Ferrite is the dominant microconstituent; however, the formation of some bainite fraction seems possible by the end of the cooling process. Water-quenching with a cooling rate of about 1000 °C/s may produce martensite formation according to the calculated TTT diagram.

*3.2. Microstructure Characterization*

Figure 3 shows a light microscope LM micrograph of the as-received steel, which indicates that the most evident microconstituents correspond to equiaxed ferrite grains with a size of approximately 30 ± 5 μm. These fine ferrite grains can be attributed to the presence of a small amount of TiC carbides. Some elongated ferrite plates are observed as part of the bainite microconstituent. No pearlite colonies seem to be present in the micrograph. This suggests that the cooling rate of about 1 °C/s was fast enough to avoid the formation of pearlite during the normalizing treatment, as indicated qualitatively in the Calphad-calculated TTT diagram. The continuous cooling transformation (CCT) diagram is more adequate for continuous cooling treatments; nevertheless, the TTT diagram was used because this diagram shows more clearly the bainitic transformation, which occurred during the cooling of the steel. It is important to note that the calculation of TTT utilizes thermodynamic data, contrary to other commercial software [15].

The carbide precipitation is not clearly observable in the optical micrograph of Figure 3. Figure 4a–f presents SEM micrographs for the steel in the as-received condition and after aging at 600 °C for 100, 300, 1000, 6000 and 7000 h. The precipitation of both intragranular and intergranular carbides is evident in the aged steel, as indicated by arrows (Figure 4b,e). This precipitation is more notorious as the aging process advances. The coarsening of carbides is evident for aging times longer than 1000 h.

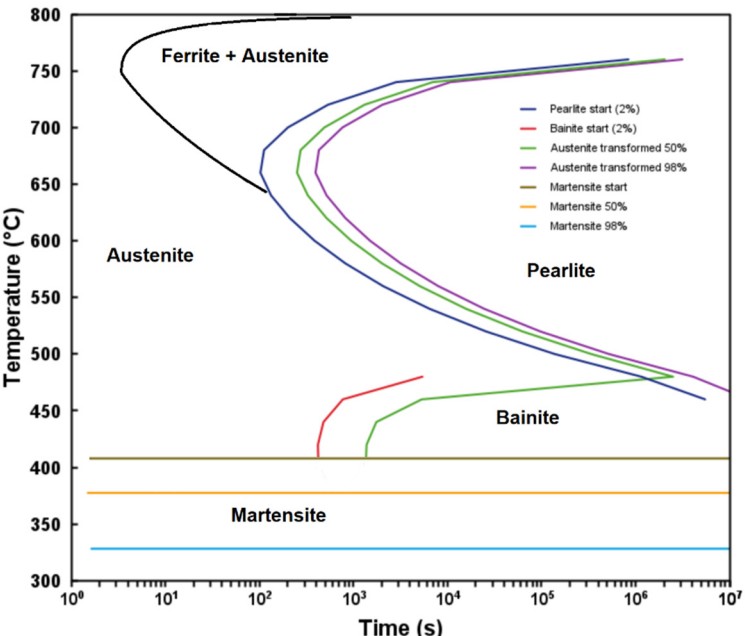

**Figure 2.** TTT diagram of the steel.

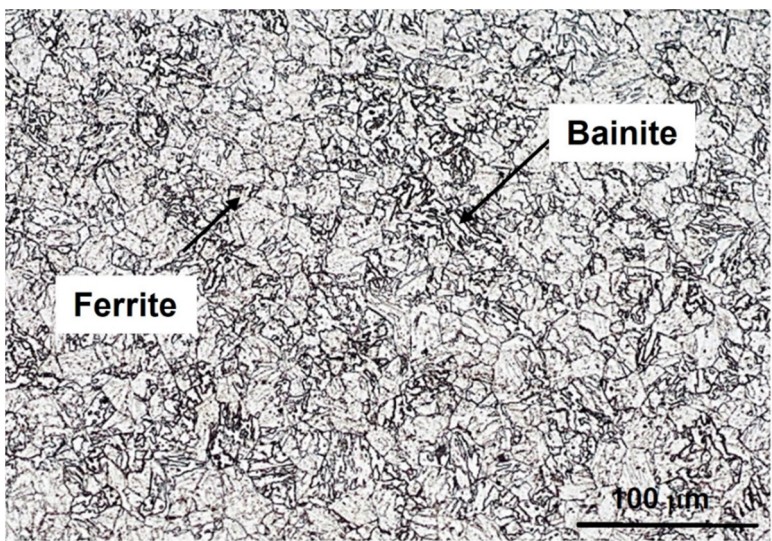

**Figure 3.** LM micrograph of the as-received steel.

Figure 5a–c presents the electrolytically extracted precipitates for the specimens after aging at 600 °C for 0 h, 300 and 7000 h, respectively. The carbide size increases with time, and the particle morphology changes from irregular to polygonal. The EDX-SEM analysis indicates that the main component for all the extracted particles is Cr.

The XRD patterns corresponding to the as-received steel and those aged at 600 °C for 100, 300, 1000 and 7000 h are presented in Figure 6. The XRD peaks of extracted precipitates from the as-received specimen are mainly of Cr-rich $M_7C_3$ carbides. The $M_7C_3$ carbide belongs to the Pmcn (62) space group [21], and the crystals present an irregular morphology, as previously presented in Figure 5a. The XRD peaks of residues from specimens aged for 300 h correspond to a mixture of $M_7C_3$ and $M_{23}C_6$ carbides. In the case of steel aged for periods longer than 300 h, the XRD patterns correspond to the $M_{23}C_6$ carbide with the Fm3 m (225) space group [22]. These carbides are faceted crystals. This suggests that the aging process promotes the change from $M_7C_3$ to $M_{23}C_6$ carbides. No $Fe_3C$ XRD peaks were detected in the diffractogram corresponding to the as-received steel, which confirms its absence, as shown in Figure 3.

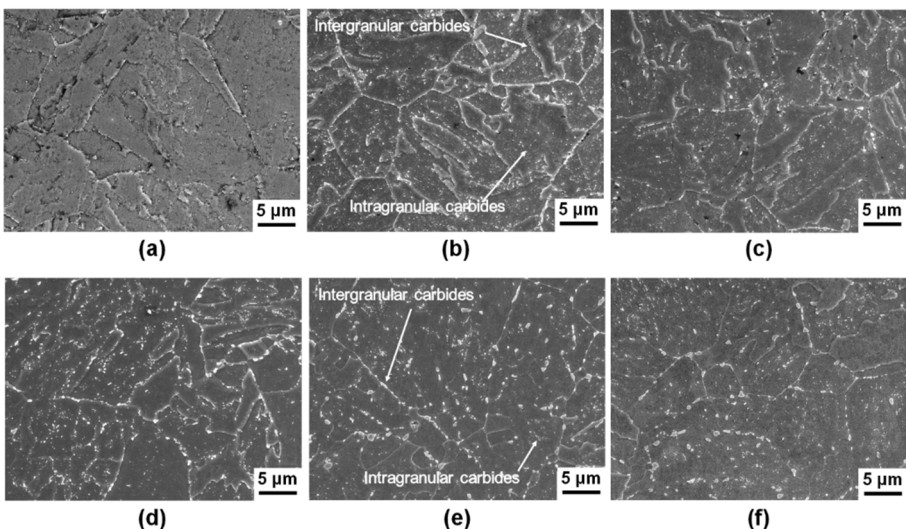

**Figure 4.** SEM micrographs of (**a**) the as-received steel and after aging at 600 °C for (**b**) 100, (**c**) 300, (**d**) 1000, (**e**) 6000 and (**f**) 7000 h.

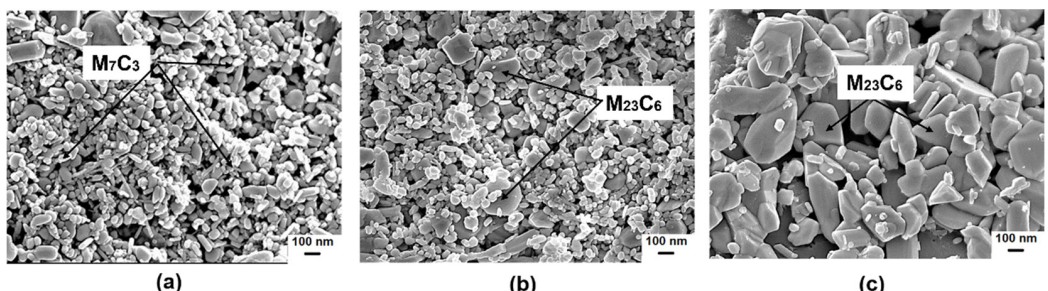

**Figure 5.** SEM micrograph of extracted residues of the steel after aging at 600 °C for (**a**) 0, (**b**) 300 and (**c**) 7000 h.

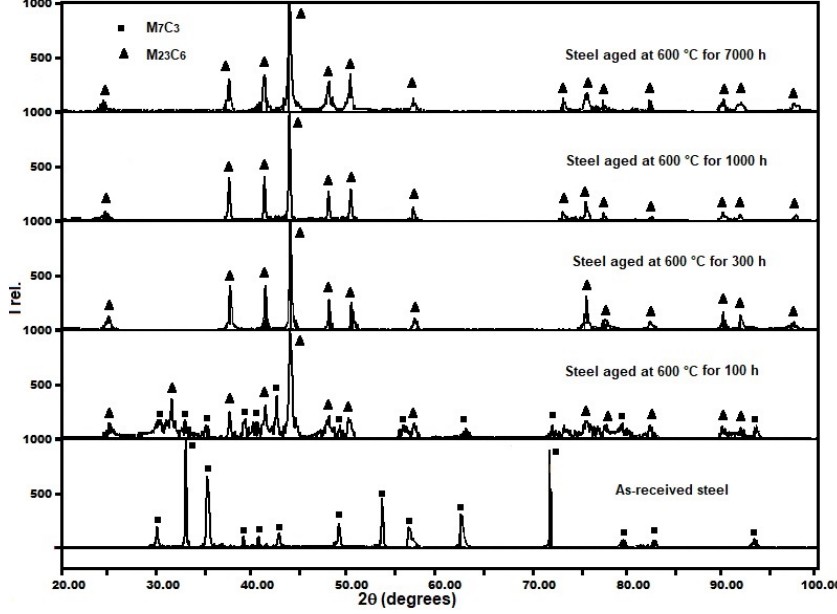

**Figure 6.** XRD patterns of the extracted residues from the steel specimen after aging at 600 °C for 0 (as-received), 100, 300 and 7000 h.

Figures 7 and 8 present the TEM micrographs and electron diffraction patterns of a precipitate extracted from the as-received steel and the steel aged for 7000 h, respectively.

Figure 7a verifies the irregular morphology of $M_7C_3$ carbide, while the electron diffraction pattern, Figure 7b, corresponds to an orthorrombic crystalline structure, Pmcn, with a zone axis ZA $[2\bar{6}1]$. Conversely, Figure 8a shows the faceted shape morphology of $M_{23}C_6$ carbides, with fcc crystalline structure, Fm3m, indicated by the electron diffraction pattern with a zone axis $[\bar{1}11]$ in Figure 8b.

**(a)**　　　　　　　　　　　　**(b)**

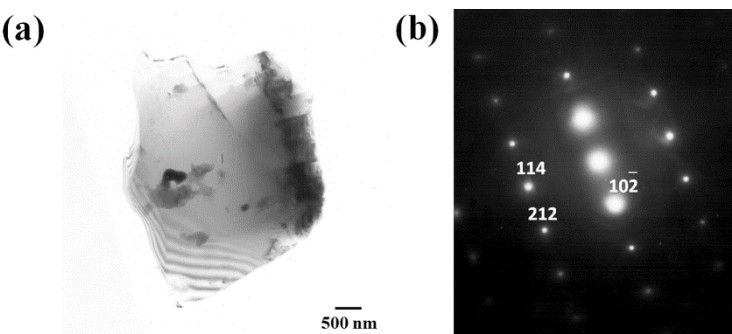

**Figure 7.** (**a**) TEM micrograph of $M_7C_3$ carbide with (**b**) electron diffraction pattern with ZA = $[\bar{1}11]$.

**(a)**　　　　　　　　　　　　**(b)**

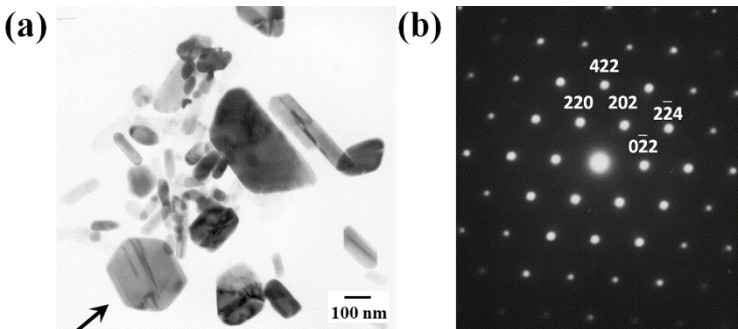

**Figure 8.** (**a**) TEM micrograph of $M_{23}C_6$ carbide with (**b**) electron diffraction pattern with ZA = $[2\bar{6}1]$.

## 4. Discussion

### 4.1. Characterization of Precipitation Evolution

The Calphad-based precipitation kinetics calculation for the normalizing process from 950 °C at a cooling rate of about 1 °C/s estimated a maximum volume fraction of $1.5 \times 10^{-10}$ for the intergranular precipitation of $M_7C_3$ carbide after the normalizing process. Neither $Fe_3C$ nor $M_{23}C_6$ precipitates were present after normalizing. The calculated precipitation kinetics of the tempering process at 715 °C for 900 s (15 min) is presented in Figure 9. The plot of volume fraction for precipitation clearly shows that the precipitation sequence starts with $Fe_3C$ formation, then $M_7C_3$ precipitation, and finally, $M_{23}C_6$ precipitation. This indicates that $Fe_3C$ is formed and subsequently disappears. This could explain the absence of cementite and the presence of $M_7C_3$ and $M_{23}C_6$ carbides in the as-received steel, as observed in Figures 6 and 7. This suggests that the precipitation reaction during tempering is as follows:

$$\alpha_{sss} \rightarrow \alpha + Fe_3C \rightarrow \alpha + M_7C_3 \rightarrow \alpha + M_{23}C_6 \tag{1}$$

On the other hand, the Calphad-calculated precipitation kinetic behavior for aging at 600 °C exhibits that the precipitation of $M_{23}C_6$ carbide is dominant, as shown in Figure 10. A very low volume fraction of $M_7C_3$ is also present during short periods, of about 1000 s, and subsequently disappears. This result agrees with the precipitated phases detected in the aged specimens by XRD and TEM in Figures 6 and 8. These results indicate that the precipitation reaction during aging at 600 °C follows the sequence:

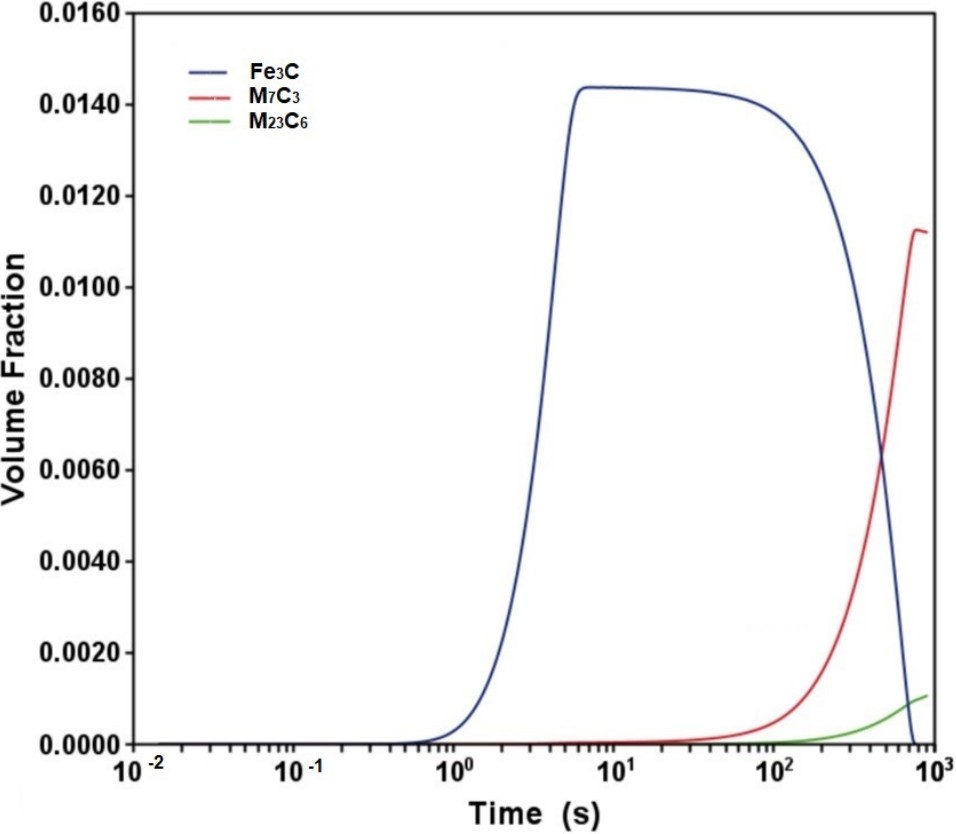

**Figure 9.** Calphad-calculated volume fraction of precipitates vs. time during aging at 715 °C.

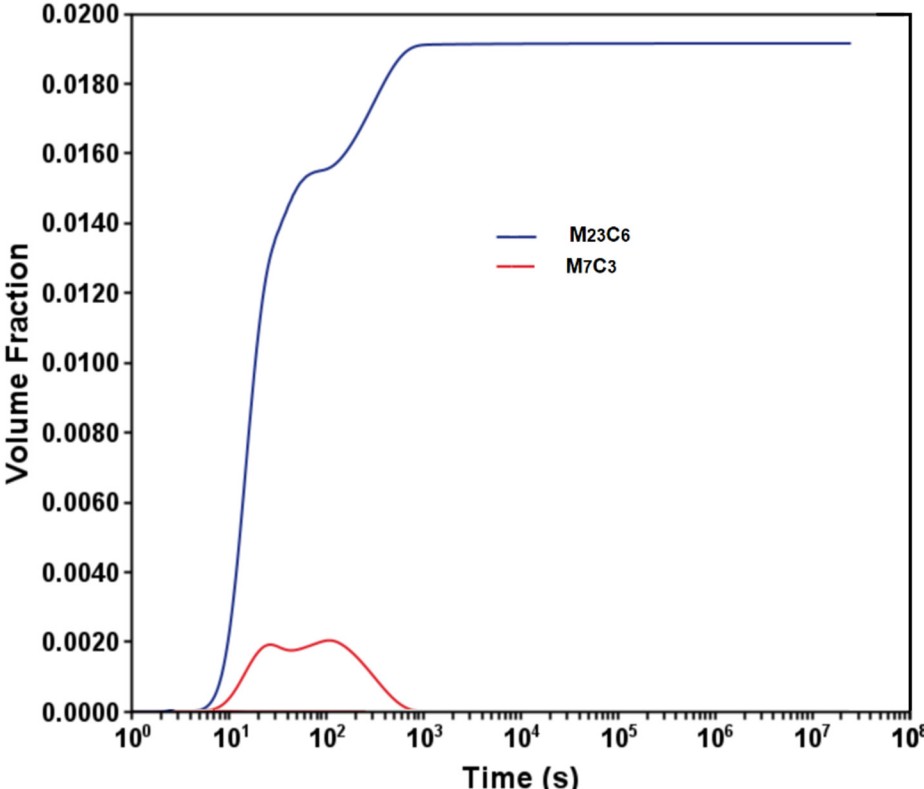

**Figure 10.** Calphad-calculated volume fraction of precipitates vs. time during aging at 600 °C.

$$\alpha_{sss} \rightarrow M_7C_3 \rightarrow \alpha + M_{23}C_6 \tag{2}$$

Figure 11 shows the Calphad-calculated precipitate mean radius against time plot with experimental values of the equivalent radius for $M_{23}C_6$ precipitates. There is a good agreement between the calculated and experimental data. The growth kinetics of precipitation for times longer than 300 h ($1.08 \times 10^6$ s) correspond to the diffusion-controlled coarsening stage. The calculated interfacial energy was 0.1 $Jm^{-2}$ between the $M_{23}C_6$ precipitate and ferrite, and 0.28 $Jm^{-2}$ between the $M_7C_3$ precipitate and ferrite. The former value suggests a coherent interface for the $M_{23}C_6$/ferrite interface [23].

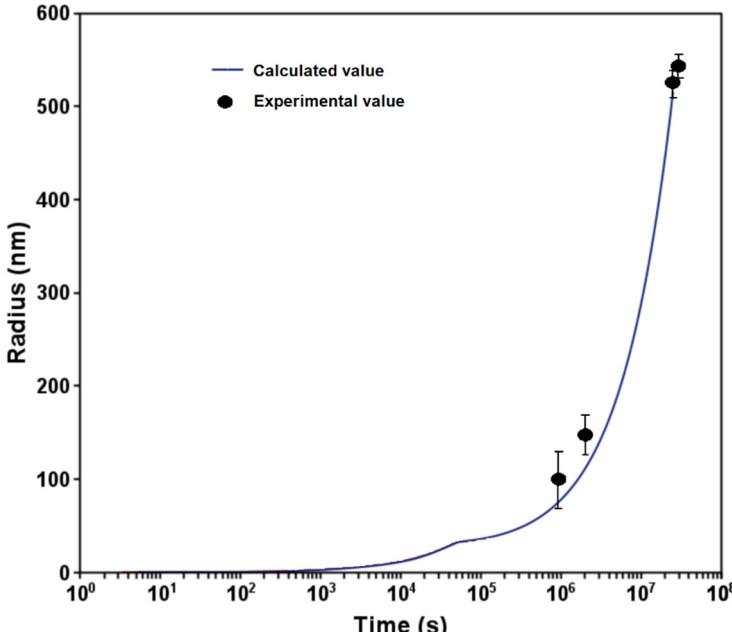

**Figure 11.** Calphad-calculated mean radius of precipitates vs. time during aging at 600 °C.

This suggests that creep strength is based on the intergranular and intragranular precipitation of Cr-rich $M_{23}C_6$, which impedes the dislocation motion and grain boundary sliding at high temperatures [1–3].

*4.2. Mechanical Characterization*

Figure 12 presents the Rockwell "B" hardness variation with aging time. There is a decreasing tendency as time proceeds. The normalizing treatment is mainly responsible for the initial hardness of 90.5 HRB due to the presence of the ferrite and bainite microconstituents.

The fast decrease in hardness from 90.5 to 86 HRB for short times can be related to the transformation of bainite into the equilibrium phases: ferrite and $M_{23}C_6$. The decrease from 86 to 83.5 HRB after prolonged periods of aging is related to the diffusion-controlled coarsening of $M_{23}C_6$ carbide, even though the precipitate size is still of nanometric order. The Calphad-calculated coarsening constant, $k$, for $M_7C_3$ carbide at 400 and 600 °C is $8.1 \times 10^{-36}$ and $1.52 \times 10^{-30}$ $m^3$ $s^{-1}$, while for $M_{23}C_6$ carbide, it is $3.8 \times 10^{-36}$ and $1.01 \times 10^{-30}$ $m^3$ $s^{-1}$, respectively. These values suggest that the coarsening resistance is higher for the $M_{23}C_6$ carbide, which is the most predominant carbide at temperatures between 400 and 600 °C, as shown by Calphad calculations. This behavior seems to be caused by the lower interfacial energy. This also indicates that the mechanical strength may remain for long aging times at these temperatures.

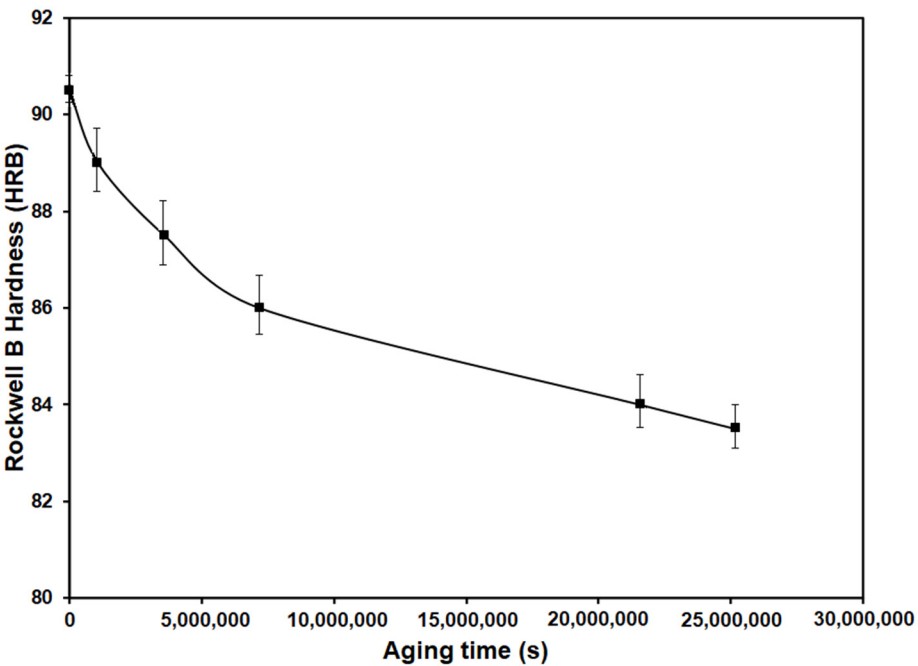

**Figure 12.** Plot of Rockwell "B" hardness vs. time for aging at 600 °C.

## 5. Conclusions

A study of phase transformations of 5Cr-0.5Mo-0.1C steel after heat treating yielded the following conclusions:

1.  The Calphad-calculated TTT diagram of the steel precisely predicted the microconstituents observed in the as-received steel.
2.  The cooling stage of the normalizing treatment promoted neither intergranular nor intragranular precipitation of carbides.
3.  The tempering process at 700 °C originated the precipitation of $M_7C_3$ and $M_{23}C_6$ carbides.
4.  Aging at 600 °C produced the precipitation of $M_{23}C_6$ carbide, which is responsible for the mechanical strength at high temperatures.
5.  The reduction in steel hardness can be attributed to the bainite transformation and diffusion-controlled coarsening of $M_{23}C_6$ carbide.

**Author Contributions:** Conceptualization, V.M.L.-H. and M.L.S.-M.; methodology, M.L.S.-M.; software, V.M.L.-H.; validation, H.J.D.-R., M.B.-Z. and D.I.R.-L.; formal analysis, M.L.S.-M. and J.M.-P.; investigation, V.M.L.-H. and M.L.S.-M.; writing—original draft preparation, V.M.L.-H., J.D.V.-C. and C.F.-P. All authors have read and agreed to the published version of the manuscript.

**Funding:** This research was funded by Conacyt, grant number A1-S-9682.

**Institutional Review Board Statement:** Not applicable.

**Informed Consent Statement:** Not applicable.

**Data Availability Statement:** The data that support the findings of this study are available from the corresponding author upon request.

**Acknowledgments:** The authors acknowledge the financial support from Conacyt A1-S-9682, and SIP-IPN.

**Conflicts of Interest:** The authors declare no conflict of interest.

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
