# Peer review of "Phase Transformations of 5Cr-0.5Mo-0.1C Steel after Heat Treatment and Isothermal Exposure"

_metals, doi:10.3390/met12081378_

Round 1

Reviewer 1 Report

1. As well known, the CCT diagram is more adequate for analyzing the phase transformation behavior during heat treatment process. Moreover, some commercial software, such as Jmatpro, can be used to simulate the CCT diagram. In line 118, the authors said: “the TTT diagram is easier to calculate”. It is necessary to give the reason for that. In addition, more experimental evidence including the initial microstructure of simulated specimens should be given in order to make the simulation more rational.

2. For the quantitative metallurgy and image analysis, such as the grain size and the mean radius of precipitates, please indicate the used methodology and investigated area including statistics.

3. The authors present that the experimental steel aged at 600 °C for times of 100, 300, 1000, 6000 and 7000 h. But in Figure 11, it can be seen that there is only two experimental values. The experimental values of mean radius of M23C6 precipitates in experimental steel under every aging time should be supplemented in order to prove the reliability of the simulation.

4.In line 65, the authors said: “ and then tempered at 715 °C for 15 min.” In line 180 and 181, the authors said: “calculed precipitation kinetics of the tempering process at 700 °C for 900 s (15 min) is present in Figure 9”. The authors should explain the reason that the heat treatment temperature of experimental steel is different from the calculated temperature.

5. XRD patterns in Figure 6 is not clear because of the strong background, especially for the steel aged at 600℃ for 100h. The authors should improve the reliability of experimental data.

Reviewer 2 Report

The manuscript entitled: Phase transformations of 5Cr-0.5Mo-0.1C steel after heat treatment and isothermal exposure aims to perform numerical and experimental analyses of the phase transformation sequence taking place in low carbon steel (5Cr-0.5Mo-0.1C steel) after heat treatment. I have the following concerns with the manuscript.

- The aim of the manuscript is not clear.

- The selection of composition 5Cr-0.5Mo-0.1C is still not clear.

- The presence of carbides is not clearly visible from both OM and SEM images. From the SEM images, the bright regions seem to be artifacts from metallographic preparation.

- Better quality XRD patterns should be introduced.

- There is no explanation as to how the carbide precipitates are extracted from the steel excepting the word electrolytically-extracted.

- Manuscript lacks a strong scientific discussion.

- Unit for the Y-axis legend is missing in Fig. 1

- Typos and the English language need attention.

Reviewer 3 Report

Dear Authors, Your Paper needs MAJOR REVISION and I strongly suggest to incorporate the below mentioned changes in Your Paper:-

1.      Include the Actual photographs of the as received steel under the section 2.2

2.      Modify the quality of the SEM Micrographs such that, the images described in Figure 4 (a) – (e) are clearly visible. Try to include HIGH RESOLUTION IMAGES as far as possible.

3.      Include Scales for the SEM Micrographs described in Figure 4(a) – (e)

4.      In Fig.5 (a) – (c), try to include some descriptions in the SEM Micrographs so that, readers can easily interpret the micro-structural transitions

   Moreover, Your submitted Paper contains a Plagiarism Level (Similarity Index) of more than 20%, which I believe exceeds the permitted level. So, I STRONGLY ADVICE the Authors to reduce the Plagiarism Level of their paper and to submit the revised paper, after checking its Plagiarism Level using Licensed Suitable Plagiarism Software.

Author Response

Please, see attached pdf file.

Round 2

Reviewer 1 Report

 the paper can be accepted.

Author Response

There is no special request for round 2.

Reviewer 2 Report

Even though the authors claim that they have revised the manuscript, I do not see any changes or any changes highlighted in the present form of the manuscript. Hence, all the questions raised in my first review report still remains open.

Author Response

Dear reviewer,

See the attached pdf file.

Reviewer 3 Report

As the Authors have modified the Paper as per the Reviewer Comments, I recommend the Paper for Publication

Author Response

(The authors gave the same response as above.)

Round 3

Reviewer 2 Report

The authors have satisfactorily addressed most of the comments and hence the manuscript may now be considered for publication.